# Reinforcement Learning Based Character Controlling

**Wang Jingbo**
Department of Information Engineering
The Chinese University of Hong Kong
Shatin, Hong Kong
wj020@ie.cuhk.edu.hk

**Yin Zijing**
Department of Information Engineering
The Chinese University of Hong Kong
Shatin, Hong Kong
yz020@ie.cuhk.edu.hk

## Abstract

Character controlling is a longstanding problem in understanding the behavior of human. This task aims to generate various and high quality human motion in the simulated environment as in the real world controlled by human. Therefore, how to use the captured real human motions is the crucial component to solve this problem. Rather than regressing the human motion directly in previous motion prediction methods, in this project, a reinforcement learning method is adopted in to our framework to learn robust control policies capable of imitating a broad range of example motion clips. Besides, we also explore the new reward functions to encourage the motion similarity between the real human and the virtual character. With these new rewards, the algorithm will convergence faster than recent advances. The representation of our project can be found in this **link**.

## 1 Introduction

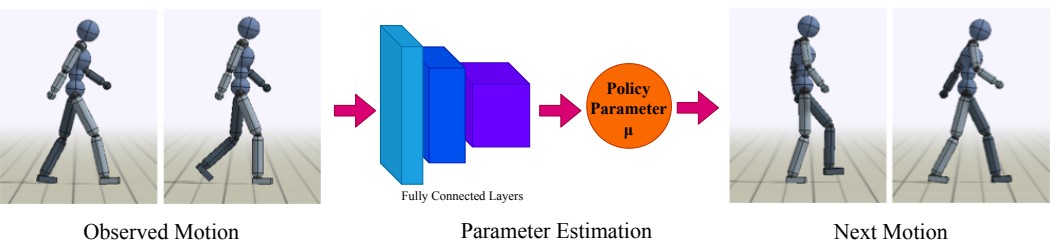

Figure 1: Framework for character controlling as motion mimicking.

Character controlling is widely utilized in many real-world applications, such as computer games, VR/AR, and motion capture. Recent methods always control what the human will do by predicting the future human motion directly, under the observation of previous human motion. However, under these methods, people cannot control the motion of virtual character in an explicit way. And thus, this problem is worth further exploration.

To solve this problem, recently, more researchers begin to model virtual character by mimicking human motions in the real world. Given human motions, the goal of this framework is to learn controllers, which have the ability to control the virtual character do same actions as the real human motions. Thus, directly, reinforcement learning, which been widely used in controlling and computer vision, can be introduced into this frame work (38; 37). Under this framework in Figure 1, the character controlling problem is changed to learn policy parameter of agents with specific predefined action, state and rewards, which can measure the differences between virtual character and real human.

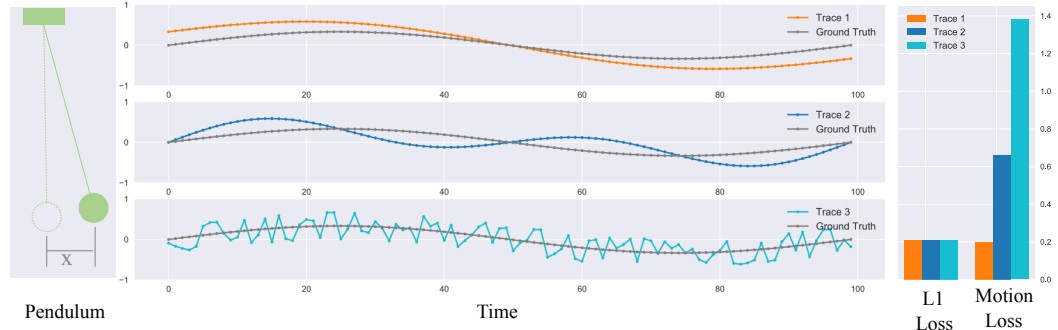

Figure 2: Motion loss is a better description of human motion.

In this project, we follow the problem definition in the well-known framework (37). Firstly, we follow the PPO optimization method which is widely used in reinforcement learning and reproduced their results which are reported in previous projects. Rather than implementing their method directly, we also do the following exploration. We also add new reward functions, which encourage the motion similarity between virtual characters and the real human motion, to speed up the Convergence process of the proposed algorithm.

## 2  Related Work

**Reinforcement Learning**   Reinforcement learning provides a feasible way for motion imitation, therefore agent can learn from trials and errors to practice human actions. Value iteration can be used in motion synthesis to develop kinematic controllers (26). However, some unnatural behaviors may occur in those manually designed controllers, such as lag and fold-over. Recently, deep neural network models have been introduced to RL for some challenging work (39) (19)(31). Some RL methods based on model-free imitation learning algorithm, such as Generative Adversarial Imitation Learning(17), can obtain significant performance in large and high-dimensional environments.

**Motion Imitation**   Motion imitation has a long history in computer animation area. One early application was designed to walk stably by following the kinematic target trajectory (41). Reference motions can be simulated to produce realistic human locomotion (25).To generate more natural human motions, reference motions have also been used in reward function (38).

**Motion Generation**   Recently, lots of work begin to focus on pose sequence generation. HP-GAN (2) combines the Seq2Seq model to the GAN framework for motion generation. Cai *et al.* (4) propose a Two-Stage GAN to generate the spatial and temporal information respectively for pose generation. PSGAN (47) takes the initial pose as input and action label as the condition to generate pose sequence for video generation. CSGN (45) formulates both generator and discriminator as graph convolution and generates pose sequence from noise sequence directly. Action2Motion (12) generates human pose sequences with a CVAE model for the given action.

**Motion Prediction**   Pose prediction is also another important task to understand human behaviors. For given continuous pose sequences, these models can predict the future human motion at a few time steps. *Encoder-Recurrent-Decoder* (ERD) (9) incorporates encoder and decoder models before and after the recurrent units for motion prediction. Based on the Seq2Seq (42) model, Martinez *et al.* (33) predicts the velocities rather than the positions of joints for motion prediction. Ac-Lstm (29) enhances the capability of LSTM by training the mixture of synthesized frames and observed frames. Graph convolution network (GCN) is also widely used in motion prediction in recent advances (7; 27; 32).

## 3  Motion Modeling

We illustrate the motion modeling of virtual characters in this section before we define the reinforcement learning framework for this character controlling task. We first define the represent the human

motion sequence and then demonstrate the method to measure the differences between prediction and target human motion.

**Human Motion** Keypoint sequence ($P_t$) is the most direct representation of human, but it ignores the rotation of human bodies and can not describe the complex human motion exactly. Thus, in this project, we add the the joint rotation sequence ($Q_t$) into the human modeling, which models the relative position of different joints directly. This rotation is always represented by quaternion, which is easily to measure the differences of angles.

**Motion differences** The direct way to measure the differences between prediction and target motion is the point to point loss, such as the distance of keypoints and the joint rotation as following:

$$L_P = \sum_{k=1}^{K} ||\hat{p_t^k} - p_t^k|| \tag{1}$$

$$L_R = \sum_{k=1}^{K} ||\hat{q_t^k} - q_t^k|| \tag{2}$$

which $\hat{p_t^k}$ and $\hat{q_t^k}$ are the predicted keypoints and joint rotation of $k$ joint at $t$ time step.

Although these two function measures the point to point differences, we also argue that this two metrics can not reflect the motion of the human exactly. As shown in Figure 2, we simplify the human motion as the pendulum motion. Although the third motion is significant worse than the first and second motion, the point-to-point loss of them of them is same. We find out the long-term motion can help to judge these motions. As shown in figure 3, we propose our motion encoding as long-term joint movement, which is encoded by the pre-defined operator and the motion difference is defined as following:

$$e_t^k = p_t^k * p_{t-T}^k \tag{3}$$

$$L_M = \sum_{k=1}^{K} ||\hat{e_t^k} - e_t^k|| \tag{4}$$

As shown in figure 2, this motion difference can represent the differences of these motion. Thus, we propose the motion metric to measure this difference and we find out that this metric is beneficial for the convergence of motion mimicking.

## 4 Experiment Setups

### 4.1 Formulation

Here we take humanoid model as an example to define states, actions and reward function, which are three major parts in RL problems.

**State** State $s$ represents the body structure of character model. Different parts of body are described as relative positions with respect to pelvis (which is the origin of coordinate system). Other configurations include its rotations and angular velocities. To be specific, x-axis is the facing direction of pelvis.

**Action** Action $a$ represents the target orientations for each joint. After sampling, the orientations are sent to PD controllers(43). Then we can get information such as moment of force and input them into physical environment.

**Reward** The origin reward is divided into two parts, namely $r_t^I$ and $r_t^G$. $r_t^G$ is designed for specific tasks such as going towards target orientations and hitting the target. $r_t^I$ encourages character to imitate reference motions and evaluates the disparity between reference data and imitations.Both of the two rewards have a weight, then we can get the total reward by adding them up.

$$r_t = w^I r_t^I + w^G r_t^G \tag{5}$$

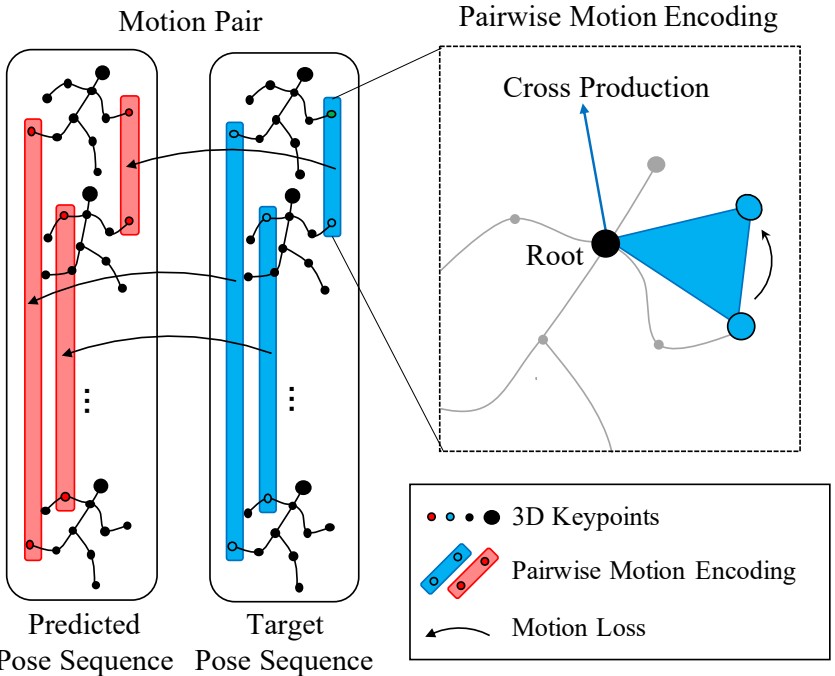

Figure 3: Illustrating of motion encoding to measure the difference between motion prediction and ground-truth.

Here, $r_t^I$ can be further decomposed as four parts, which represent pose, velocity, end-effector and center-of-mass respectively. At each step $t$, $r_t^I$ calculates imitation reward according to these parameters.

$$r_t^I = w^p r_t^p + w^v r_t^v + w^e r_t^e + w^c r_t^c \qquad (6)$$

In these rewards, the formulation of $r_t^p$ and $r_t^e$ are same as $L_R$ and $L_P$ respectively.

Besides, we also adopt the motion metric as the new reward function when the time step is larger than the predefined motion range $T$. Thus, this reward function can be written as following:

$$r_t^I = w^p r_t^p + w^v r_t^v + w^e r_t^e + w^c r_t^c + \mathbf{1}(\mathbf{t} >= \mathbf{T}) w^m r_t^m \qquad (7)$$

Although $r_v$ is in this function, we argue that the effectiveness of the short-term motion is minor for speeding up convergence of this framework. We will evaluate the effectiveness of the new reward function can help the reinforcement in the following experiments.

## 4.2  Training algorithm

Policy gradient methods are fundamental in deep neural networks and reinforcement learning, but they are sensitive to step size and sometimes have poor efficiency. To solve these problems, Proximal Policy Optimization (PPO) was proposed as a simple but useful off-policy algorithm to train an agent. It has a new objective function to achieve small batch updates in multiple training steps, which can minimize the cost function and reduce the deviation from previous policy. In this algorithm, $r_t(\theta)$ denotes the probability ratio under the new and old policies, which is represented as follows:

$$r_t(\theta) = \frac{\pi_\theta(a_t|s_t)}{\pi_{\theta_{old}}(a_t|s_t)} \qquad (8)$$

In order to make the training more stable, the advantage function will be clipped if the probability ratio between the new and old policies. Then the final objective function can be represented as follows:

$$L_t(\theta) = min(r_t(\theta)\hat{A}_t, clip(r_t(\theta), 1 - \epsilon, 1 + \epsilon)\hat{A}_t) \qquad (9)$$

Where $\hat{A}_t$ is the estimated advantage at time $t$, and $\epsilon$ is usually set to 0.2.

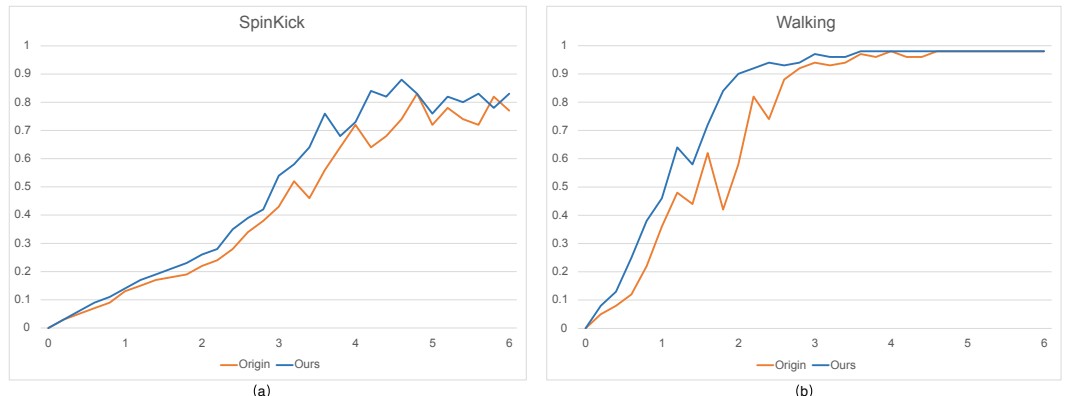

Figure 4: **Visualization result of learning curve** . The orange curves represent the learning process with origin reward function and the blue curves are for the reward with motion metric. Our method speeds up the convergence of the previous proposed framework.

# 5   Experiments

This section, we first illustrate the experiment settings of our project. Then, we evaluate our method can speed up the convergence of the previous reinforcement learning framework. At last, we will demonstrate several visualization results of the motion mimicking framework.

**Experiment Settings**   We conduct all experiments based on *Tensorflow Toolbox* and the visualization results are based on *OpenGL*. All the reference human motions are provided by the DeepMimic Project (37) and we only focus on the mimicking the single motion of human character. Especially, in our project, we train our model on the hard action "SpinKick"and the easy action"Walking" for convenience to evaluate that our new reward function can help convergence of this framework. Besides, as (37), we add the negative exponential function to all motion metrics and follows the weights of these rewards.

**Experiments Results**   We demonstrate the learning curves of "SpinKick" and "Walking" actions under the origin and proposed reward functions in Figure 4. Although both rewards can make the framework converge to a stable situation, our reward function can help this system converge faster than the system ignored long-term motion metric. Especially for the easy action " Walking", which just contains several simple motion patterns, the framework achieves the stable solution faster than the origin one significantly.

**Quantitative Results**   At last, we will show how this model works by visualizing the action of controlled virtual character. For convenience, we just visualized the hard action "SpinKick" in the following figure 5. This figure show the "SpinKick" action controlled by the trained model and we think this action is very similar to the human action in the real world.

# 6   Conclusion

Character controlling is a difficult and long standing problem for computer vision and reinforcement learning. And the reward function is crucial for this system to learn accurate human motion efficiently. In this project, we first analyze the metric to measure the differences between the motion prediction and ground-truth. Then, based on the reproduced motion mimicking framework under the reinforcement learning. We find out that the motion reward can speed up the convergence of the complex system. At last, we visualize the final results of the virtual character who can do different actions as the reference humans. We believe that with more accurate motion description and efficient reinforcement learning framework, we model the virtual do more feasible, complex, and challenging actions as human.

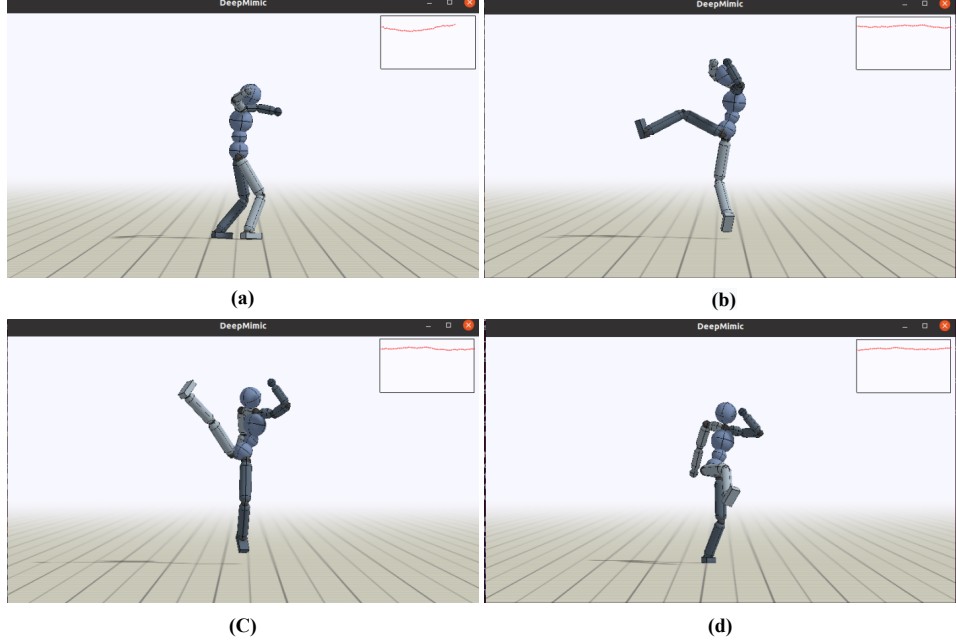

Figure 5: **Visualization result**. This reinforcement learning framework can control the virtual character do hard action similar with the human in real worlds.

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
