# OpenReview forum: "Reinforcement Learning Based Character Controlling"
_CUHK.edu.hk/2021/Course/IERG5350_

### Official Review · AnonReviewer3 · 2020-12-16
**Propose new loss and reward function for PPO in character controlling proplem.**

**Rating:** 7
**Confidence:** 3

**Review:**

General:
This paper studies the character controlling problem. They analysis the original point2point L1 loss and find it can't reflect the motion of the human exactly. So they propose another motion loss and also introduce new reward function to help boost the training process and performance.

Innovation:
Figure 2 and 3 are important to help us understand the shortage of the original L1 Loss and follow authors' idea to propose Motion Loss.
I think the proposal of this new loss and reward function is the key innovation of this project.

Comments:
1.Technical quality: Clear defined formula and methodology.
2.Clarity: The paper is well-organized and easy to read. Figures you provided help us better understand your explanation and experiment.
3. Hope you can do more experiments to explore in this projects.

---

### Official Review · AnonReviewer2 · 2020-12-20
**Good work with novelty.**

**Rating:** 8
**Confidence:** 4

**Review:**

# Summary
The authors proposed a new reward definition that can improve the convergence speed of the RL algorithm in the Character Controlling environment.

# Novelty
The work of this paper is novel, as it proposed a new better reward definition based on the authors' understanding of the environment.

# Significance
This paper made contributions to this research area, which is applied in many real-life scenarios. Therefore, I think this work is significant.

# Strong Points
1) The authors have a deep understanding of the Character Controlling Problem, thus give a clear introduction and explanation of their work.
2) The proposed reward function is novel and inspiring.
3) The paper is well organized and written.

# Weak Points
1) I think the author could try more RL algorithms to check whether their proposed reward function has a general improvement on all RL algorithms.
2) Some equations can be written better. E.g. In equation (3), what does T means.

# Revision Advice
1) Improve the expression of your equations, such as equation (3) and (7)
2) Try more algorithms to see the functionality of the proposed reward.